# Plasmid streamlining drives the extinction of antibiotic resistance plasmids under selection for horizontal transmission

Andrew C. Matthews[1], Sonja Lehtinen[2], Tatiana Dimitriu ®[3]*

1 Environment and Sustainability Institute, University of Exeter, Penryn, United Kingdom, 2 Department of Computational Biology, University of Lausanne, Lausanne, Switzerland, 3 Biomedical Sciences Research Complex, School of Biology, University of St Andrews, St Andrews, United Kingdom

* td79@st-andrews.ac.uk

## Abstract

Conjugative plasmids carrying antimicrobial resistance (AMR) genes are critical for the spread of AMR, due to their ability to transmit horizontally between bacterial hosts. We previously observed that during experimental evolution in the presence of abundant susceptible *Escherichia coli* hosts, the AMR plasmid R1 rapidly evolves variants with increased horizontal transmission due to mutations causing increased plasmid copy number. Yet AMR was progressively lost from the evolving populations. Here, we show that AMR loss was associated with evolution of streamlined plasmids in which the AMR region is spontaneously deleted, making plasmid carriage undetectable by plating on selective antibiotic-containing media. These plasmids transmit both vertically and horizontally more efficiently than the ancestral AMR plasmid, driving AMR extinction in bacterial populations and effectively acting as an intrinsic defence against AMR plasmids. A simple model of plasmid competition further shows that any horizontal or vertical transmission advantage conferred by plasmid streamlining would be enough to drive the displacement of competing AMR plasmids, with a given horizontal transmission advantage leading to faster replacement in conditions favoring horizontal transmission. Our results suggest that within-host plasmid evolution or engineered streamlined plasmids could be exploited to limit the spread of AMR in natural populations of bacteria.

## Introduction

In bacteria, conjugative plasmids are mobile genetic elements with the ability to transmit horizontally from a donor to a recipient cell, both within and between species. Plasmids play a central role in the dissemination of antimicrobial resistance (AMR) genes among pathogenic bacteria [1,2]: a few major plasmid lineages are responsible for the spread of clinically relevant AMR genes, including carbapenamases and extended-spectrum β-lactamases, among gram-negative bacteria [3]. Horizontal transmission via conjugation has specifically been implicated in the dissemination

**Data availability statement:** All relevant data are within the paper and its Supporting information files.

**Funding:** T.D. acknowledges funding support from the Royal Society (University Research Fellowship URF\R1\231740) https://royalsociety.org/. The funders played no role in the study design, data collection and analysis, decision to publish, or preparation of the manuscript.

**Competing interests:** The authors have declared that no competing interests exist.

**Abbreviations :** AMR, antimicrobial resistance; DAP, diaminopimelic acid; IS, insertion sequences; PCN, plasmid copy number; RM, restriction-modification.

of AMR. For instance, worldwide dissemination of the gene *mcr-1* encoding colistin resistance is due to the transmission of a few promiscuous plasmids between strains [4]; and transfer of an azithromycin-resistant plasmid facilitated epidemics across multiple *Shigella* species [5]. Plasmids can also be responsible for the transmission of AMR genes between strains living in different environments, e.g., farm animals and humans [6]. At the other end of the scale, plasmid transmission occurs between species of clinical enterobacteria within the gut of hospitalized patients [7]. Thus, it is crucial to understand what drives horizontal transmission of conjugative AMR plasmids, and what barriers exist to transmission.

Plasmid conjugation depends on both environmental factors (e.g., temperature or spatial structure) and genetic factors [1]. It is primarily controlled by the expression of the plasmid-encoded conjugation machinery and its regulatory network [8], but both donor and recipient genotypes also impact conjugation [9], as well as their relatedness [10]. Defence systems present in recipients, although likely to have evolved mostly in response to phage predation, also impact conjugation [11], and can in turn impact the distribution of AMR genes in pathogens [12]. Finally, plasmids themselves can exclude other plasmids, via surface or entry exclusion [13], by competition for replication (incompatibility [14]) or by encoding defence systems targeting plasmids [15].

We previously studied experimentally the short-term evolution of R1, a model conjugative plasmid conferring resistance to multiple antibiotics, and one of the first and best studied conjugative plasmids [16,17]. It belongs to the F-like plasmids, widespread in Enterobacteriaceae and commonly responsible for AMR [18,19]. In structure and regulation of the transfer operon, R1 is representative of the largest group of F-like plasmids [20]. In our previous study, we asked how R1 horizontal transmission evolves in the absence of antibiotic selection, and in the presence or absence of potential recipients [21] (Fig 1A). In the presence of potential recipients providing selection for horizontal transmission, plasmids with increased conjugation rate rapidly evolved [21]. In most clones, this was due to mutations within the *copA* gene controlling plasmid replication, associated with an increase in R1 plasmid copy number (PCN). Despite the increased conjugation rate of evolved AMR plasmids, at the population level, there was progressive extinction of AMR in the treatments with larger daily influx of recipient cells. We interpreted this as a decline in plasmid-carrying lineages, due to plasmid horizontal transmission being too low for plasmids to maintain themselves when facing repeated influx of plasmid-free cells. Here, we discover that instead, this behavior was driven by the rapid evolution and invasion of streamlined plasmids characterized by deletion of a large region containing all of the plasmid AMR genes. We show that streamlined plasmids can displace the ancestral AMR plasmids both vertically and horizontally, and act as an effective barrier against AMR plasmids within bacterial populations. The vertical transmission advantage is linked to higher PCN in evolved R1 variants. By contrast, the horizontal transmission advantage appears to be associated with the shorter length of streamlined plasmids, and may be generalizable to other plasmid types. A model of plasmid competition further shows that any horizontal or vertical transmission advantage is sufficient for displacing competing plasmid variants, and that displacement will happen faster in conditions that generally promote plasmid transmission.

## Results

### Existence of streamlined plasmids with deletion of the AMR region within AMR clones

Previously, we studied R1 plasmid evolution when present in either wild-type (w) or mutator (m) *Escherichia coli* hosts, and in the presence of either none or increasing proportions of immigrant plasmid-free cells (conditions designed to favor horizontal transmission) ([21], Fig 1A). We isolated one plasmid-carrying clone per evolved population (see nomenclature in Fig 1A), by plating evolved populations in the presence of ampicillin. Short-read sequencing identified multiple plasmid variants carrying point mutations in the *copA* gene (summarized as *copA** mutations), associated to an increased PCN,

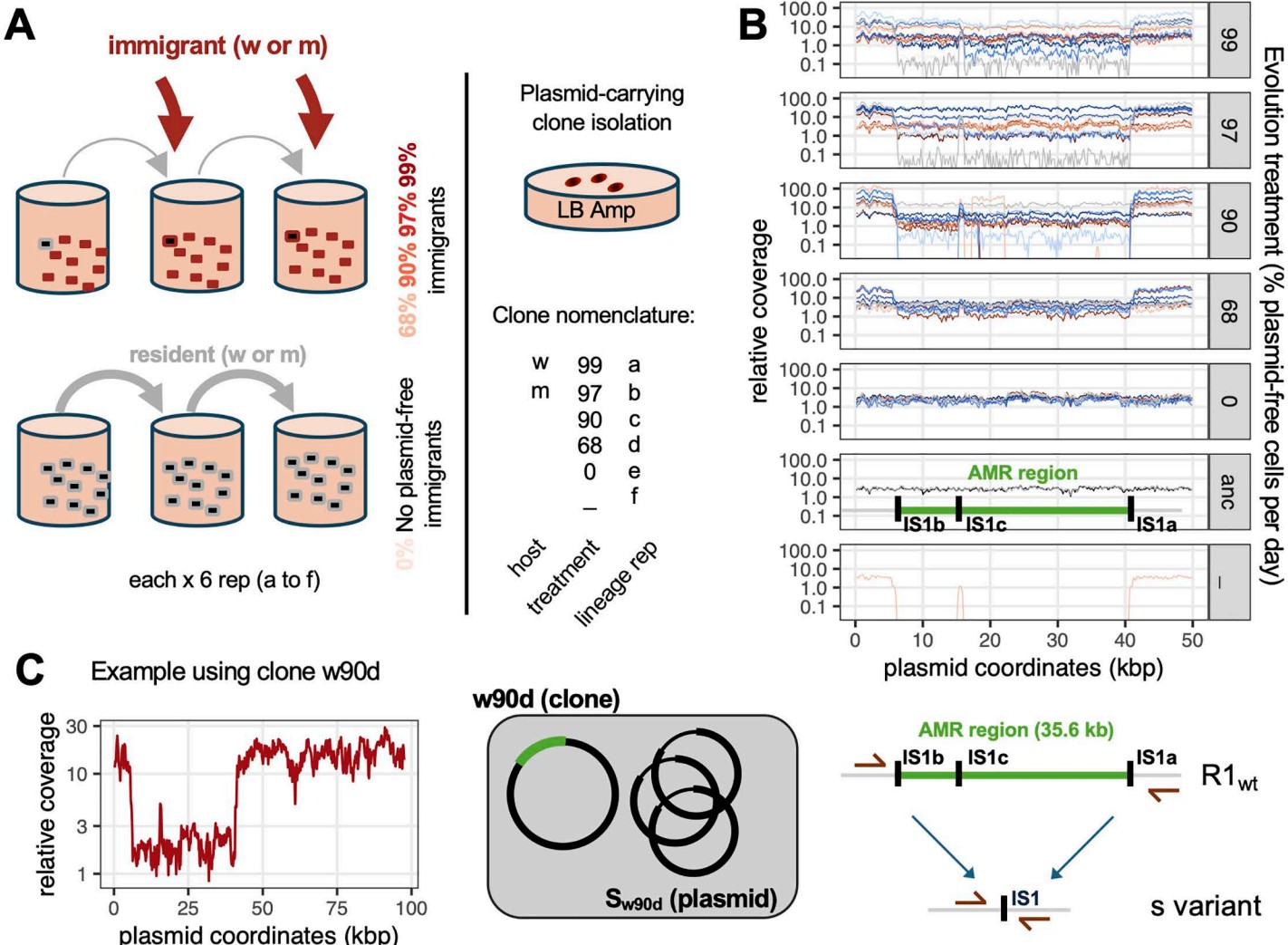

**Fig 1. Evolved plasmid-carrying clones with high copy number contain a mix of full-length and streamlined plasmids. A**: summary of the initial experimental evolution design and nomenclature used for the evolved plasmid-carrying clones. Host strains had either wildtype (w) or increased (m) mutation rate. Treatment indicates the % of plasmid-free hosts added at each passage during evolution. **B**: coverage map of evolved clones. Relative coverage of sequencing reads is shown for all clones across R1 sequence length (only the first 50 kb are shown). The AMR region and copies of *IS*1 are shown against the ancestral R1 ("anc" treatment) coverage map. **C**: schematics of the streamlined variant mutations. On the left are shown relative coverage and interpretation of plasmid content for sequenced clone w90d; on the right a map of the deletion of the AMR region between external *IS*1 sequences is shown, with the position of primers used for screening and sequencing. The data underlying this figure can be found in S1 Data.

which caused increased horizontal transmission via gene dosage effects [21]. Increased PCN of *copA** plasmids was apparent as increased coverage of mapped reads when mapping reads from Illumina sequencing to the ancestral plasmid sequence. However, we also noticed that the depth of coverage of sequencing reads was not uniformly high along the length of R1 plasmid sequence: instead, for many clones there was a drop in sequencing depth approximately between coordinates 5,700 and 41,300 (Figs 1B and S1). Depending on the clones, relative coverage in this region was either higher than the one of the ancestral plasmid, equal to the ancestral plasmid (2–3 copies per chromosome), or lower than the ancestral plasmid or even than the chromosome (Fig 1B). We hypothesized that this variation in sequencing depth was due to a large deletion (approximately 35.6 kb long) present within R1 sequence, in some of the plasmid copies, and aimed to characterize this potential deletion in more detail.

Strikingly, the low coverage region included the AMR region containing all antibiotic resistance genes present on R1, including the *bla* gene conferring ampicillin resistance. As we isolated R1-carrying clones using ampicillin, the clones we sequenced could not have fully lost the AMR region, and the result is a mixture of R1 sequence variants co-existing within each heterogeneous population emerging from the single isolated clone (likely facilitated by the evolution of high copy number). To identify and characterize these plasmid variants, we took advantage of control populations from our evolution experiment which were established without plasmid, and for which we had sequenced clones without any antibiotic selection step. In one clone, w_e, sequencing revealed the existence of reads mapping to R1, which must have originated from contamination (likely from the plasmid-carrying populations that were evolving nearby in the same 96-well plate—see Supplementary Text in ref [21] for further discussion). In clone w_e, no reads mapped within the AMR region, suggesting a full deletion (Fig 1B bottom). We designed PCR primers around the AMR region (~35.6 kb long in ancestral R1), which yielded a product just above 1 kb long in clone w_e, confirming a large deletion exists within the evolved plasmid. Clones carrying the ancestral R1 plasmid also yielded a faint band of the same size, but control experiments showed that this was due to in vitro events and not in vivo recombination (see Materials and methods and S2 Fig). Differences in band intensity still allowed us to reliably identify presence of the deletion.

Plasmids with deletion of the AMR region retain the plasmid 'backbone' including all functions essential for plasmid maintenance and conjugation. For this reason, we refer to plasmids with a deletion of the AMR region as streamlined plasmids. We distinguish between bacterial clones (that can contain a mixture of plasmids) and variant plasmids by identifying streamlined plasmids with the notation s. For instance, $s_{w\_e}$ is the streamlined plasmid variant present in clone w_e, and $s_{w90d}$ is the streamlined plasmid variant present in clone $w_{90d}$, where coverage data suggests it co-exists with a full-length AMR variant (Fig 1C). The deleted region is bounded in the ancestral R1 plasmid by two identical insertion sequences (IS) IS1 copies, IS1b and IS1a, in direct orientation. Sanger sequencing revealed that $s_{w\_e}$ variant experienced a deletion bounded exactly by the two external IS sequences IS1b and IS1a. It retains one unique IS copy, which suggests this variant evolved by homologous recombination between the two initial direct repeats (Fig 1C), and not by transposase-mediated deletion.

Coverage data showed uneven read coverage along the plasmid sequence for many evolved clones selected using ampicillin, corresponding to the co-existence of streamlined and full-length plasmids (Fig 1B). To quantify this, we compared short-read coverage between the AMR region and the rest of the plasmid, or plasmid backbone (S3A Fig). While a few evolved clones had similar coverage between the AMR region and the plasmid backbone, the majority of high PCN clones had higher coverage for the backbone, with many of them showing no increased coverage for the AMR region compared to the ancestor. This suggests that the majority of these clones contained one version of the plasmid similar to the ancestor $R1_{wt}$, together with a high copy number, streamlined plasmid. Accordingly, the large majority of clones in which the *copA** allele was detected had a *copA** allele frequency of around 0.9 and a relative frequency of the AMR region (when compared to the backbone) of around 0.1 (S3B Fig). *copA** allele frequency and AMR region coverage were negatively, linearly correlated across clones, suggesting that most clones carry two types of plasmid molecules: a streamlined, high PCN *copA** variant and a full-length variant with ancestral, low PCN.

 

## Extinction of AMR is associated with the spread of streamlined plasmids

We previously followed plasmid population dynamics using the ampicillin resistance phenotype conferred by the *bla* gene within the AMR region in the full-length plasmid and concluded that plasmids went extinct [21]. Identification of streamlined variants in sequenced clones suggests that instead, AMR plasmids went extinct but streamlined plasmids might have survived for longer and escaped detection in antibiotic-sensitive clones. Consistent with AMR plasmids being subject to evolutionary rather than direct ecological dynamics is the observation that AMR population density began declining only after 12–15 days. If ecological dynamics alone drove this decline (due to insufficient horizontal transmission), we would expect a steady trend beginning much earlier. To test the robustness of this delayed AMR decline, we first repeated a subset of our previous experiment (adding either no recipient or plasmid-free recipients representing 95% of the total bacterial population every day) for 32 days. We found that the population dynamics in our repeat evolution experiment was very similar to the first (Fig 2A) with AMR cell density beginning to decline below its initial levels after approximately 10 days in the presence of plasmid-free recipients. Interestingly, this time we observed a later decline in AMR cell density even in the absence of plasmid-free recipients.

In light of our results on individual clones, we revisited our analysis of plasmid population dynamics data. We hypothesized that streamlined plasmids might have persisted for the full length of the experiment but escaped detection using antibiotic resistance as a marker, due to deletion of antibiotic resistance determinants. We used PCR with the R1$_{del}$ primer pair to discriminate between full-length and streamlined plasmid carriage (see S2 Fig), and screened colonies obtained from reviving frozen endpoint populations on LB-agar in the absence of antibiotic selection (Fig 2B). Strikingly, we found that streamlined R1 variants were present in most colonies tested for all populations evolved with regular influx of plasmid-free cells. In the second experiment, in which we observed a late decline in AMR cell density even in treatments without regular influx of plasmid-free cells,

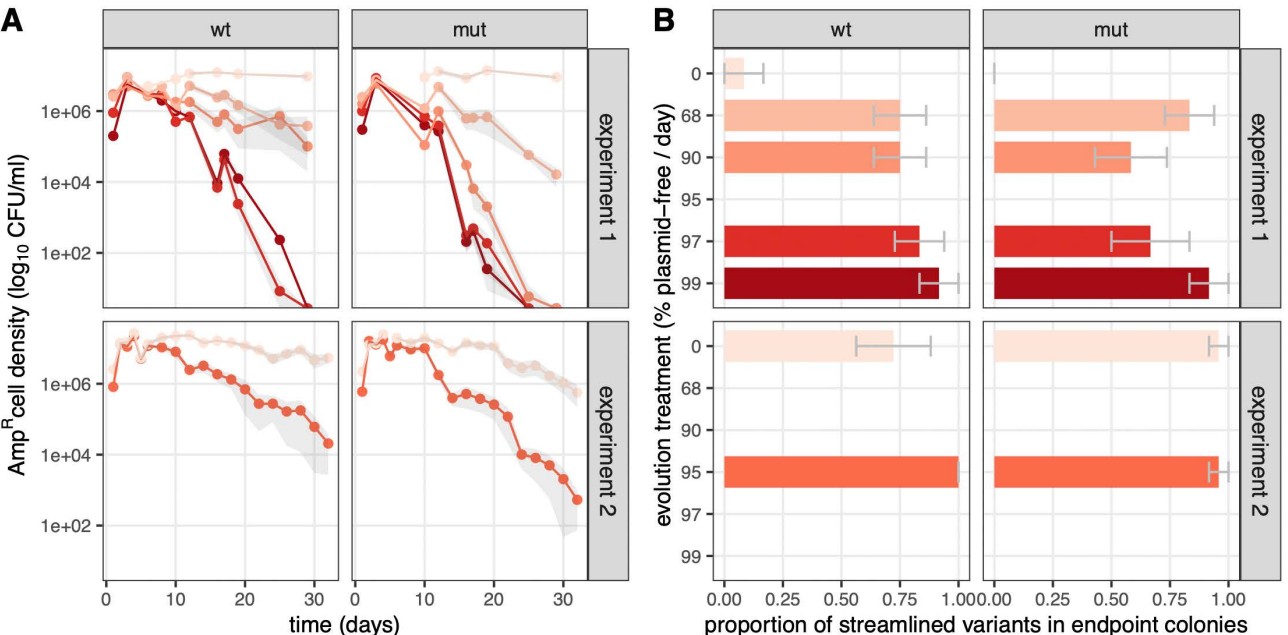

**Fig 2. AMR declines repeatedly during evolution in the presence of plasmid-free host bacteria, but streamlined plasmids are maintained at high density. A** shows the dynamics of ampicillin resistant cell density in two independent experiments for two host backgrounds (wt and mut). **B** shows endpoint proportions of streamlined R1 variant carriage. Color and y-axis in B indicate evolution treatment. Each treatment × host background combination was run in 6 independent evolution replicates; mean values per conditions are shown in color and mean ± s.e.m. is shown in gray, as a shaded area in A and as an error bar in B. The data underlying this figure can be found in S1 Data.

endpoint populations for these treatments also contained a large proportion of streamlined R1 variants. Thus, the observed decline in AMR cell density was not due to plasmids being lost from host populations; instead streamlined variant plasmids spread through host populations. This suggests that the decline in AMR full-length plasmids might be itself due to displacement by streamlined variants. To understand why streamlined plasmids displace full-length AMR plasmids at the population level, we next studied how streamlined plasmids impact host growth, and AMR plasmid transmission within host populations.

### Both full-length and streamlined plasmids are stable when replicating independently

As a preliminary experiment, we confirmed that in the absence of plasmid competition, different plasmid variants (R1, low PCN $s_{w\_e}$ and high PCN $s_{w90d}$) are stably maintained: over 8 days, no plasmid loss was detected (S4 Fig). This is expected due to the diverse stability mechanisms present on R1, that ensure very low rates of plasmid loss [22]. This result implies that over this timescale, plasmid loss is minimal. Due to entry exclusion between plasmids [13], horizontal transmission should in turn be negligible in experiments where all bacterial cells initially carry plasmids. Furthermore, we observed no streamlined variants in the R1 populations. This suggests that timescales up to 8 days are appropriate to compare variant population dynamics, which are expected to be primarily driven by competition rather than evolutionary dynamics.

Next, we studied how plasmid variants differ in their cost to the host, as well as vertical and horizontal transmission. We focused on a subset of variants differing in AMR region presence as well as their *copA* and *finO* alleles (S1 Table), as we expected these to impact cost and/or transmission: *copA** variants have increased PCN, and inactivation of *finO* leads to derepression of the transfer operon and a more than 1000-fold increase in R1 transfer [23]. The variants we used included several streamlined plasmids, as well as m97e mentioned above, and R1*finO*, a full-length R1 variant characterized in [21] which carries a single point mutation inactivating the *finO* gene.

### *finO* and *copA* genotypes have more impact on host growth than plasmid streamlining

To measure how plasmid variants affect the growth of their hosts, we measured the exponential growth rate of two strains (MG Rif$^R$ and MG Δ*lac*, both used as standard plasmid hosts for transmission experiments below) alone or carrying R1 or one of R1's evolved variants (Figs 3 and S5 for detail of growth curves and replicate data). As we observed significant variation in exponential growth rate depending on the overnight culture (see S5B Fig), all experiments were run using three independent overnight cultures. Overall, MG Rif$^R$ grew more slowly than MG Δ*lac*, and plasmid type had a significant effect on growth rate, which also depended on the host (growth rate ~ strain * plasmid * replicate culture, strain effect $F_{1,518} = 1,917$, $p < 2 \times 10^{-16}$, plasmid effect $F_{7,518} = 443$, $p < 2 \times 10^{-16}$, interaction effect $F_{7,518} = 121$, $p < 2 \times 10^{-16}$).

Neither the ancestral R1 plasmid or the low PCN $s_{w\_e}$ had a detectable effect on growth rate (see S3 Table for detailed statistics). To understand if the effect of AMR region deletion depends on PCN, we included m97e plasmid, one of the three evolved R1 variants that had uniformly high PCN (with no AMR region deletion). m97e's cost on growth was significant (TukeyHSD test versus plasmid-free, $-0.04 \pm 0.02\,h^{-1}$, $p = 3 \times 10^{-10}$), and significantly higher than R1 cost (TukeyHSD test versus R1, $-0.03 \pm 0.02\,h^{-1}$, $p = 6 \times 10^{-6}$). Interestingly, strains carrying either of two high PCN streamlined variants tested, $s_{w90d}$ and $s_{m97f}$, had no significant cost, and grew significantly faster than m97e- carrying strains (TukeyHSD test versus m97e, $0.035 \pm 0.02\,h^{-1}$, $p = 6 \times 10^{-7}$ for $s_{w90d}$, $0.04 \pm 0.02\,h^{-1}$, $p = 4 \times 10^{-9}$ for $s_{m97f}$), showing that deletion of the AMR region alleviates the cost of this region when present on high PCN plasmids. Finally, we also measured the cost of the R1$_{finO}$ variant which displays derepressed transfer, as well as the cost of $s_{m68c}$, a streamlined variant which has both high PCN and a *finO* mutation identical to the one in R1$_{finO}$. We found that R1$_{finO}$ also imposed a significant and large cost ($-0.11 \pm 0.02\,h^{-1}$ versus plasmid-free clone, $p = 2 \times 10^{-10}$); and $s_{m68c}$ imposed a much higher cost again ($-0.26 \pm 0.02\,h^{-1}$ versus plasmid-free clone, $p = 2 \times 10^{-10}$, $-0.14 \pm 0.02\,h^{-1}$ versus R1-carrying clone, $p = 2 \times 10^{-10}$). This is consistent with transfer gene expression placing a high burden on the host, as high PCN will multiply the effect of increased gene expression from *finO* mutations. Thus, overall plasmid carriage cost appears to be determined first by the level of transfer gene expression, followed by PCN, and deletion of the AMR region ameliorates plasmid cost significantly specifically in high PCN plasmids.

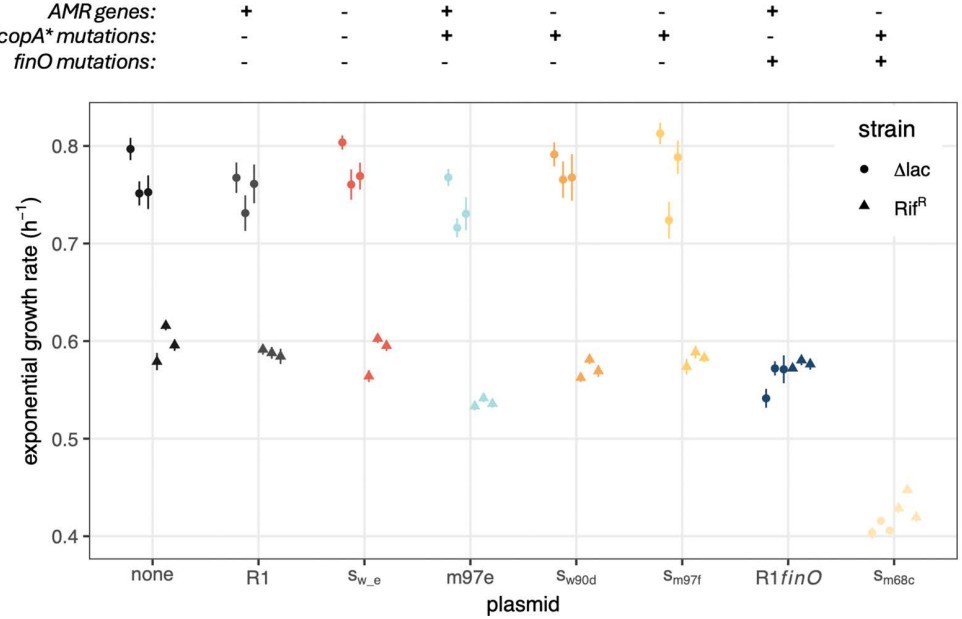

**Fig 3. Plasmid cost driven by copy number and transfer genotypes is partially alleviated by AMR gene deletion.** The effect of plasmid variants on exponential growth rate was measured in two strain backgrounds, MG Δ*lac* (dots) and MG Rif$^R$ (triangles). Each dot corresponds to the average of 12 technical replicates for an independently grown overnight culture (*n* = 3 per strain, 6 in total); lines represent the average ± SE, calculated across the 12 technical replicates. Key plasmid genotypes are shown at the top. The data underlying this figure can be found in S1 Data.

### Streamlined plasmid variants transmit more efficiently than AMR plasmids

To understand how streamlined plasmids affect AMR plasmid vertical transmission, we followed R1-carrying cell density (using its ampicillin resistance phenotype) in clones where it co-exists with a streamlined variant at the start of the experiment. Here, we focused on two streamlined variants varying in their PCN: $s_{w\_e}$, which has an ancestral, low PCN, and $s_{w90d}$, which evolved high PCN due to a G589T mutation in *copA*, making it a representative of a common *copA\** genotype among evolved plasmids [21]. Over 8 days, R1 was maintained in the population with no detectable loss when competing with the low PCN $s_{w\_e}$. In contrast, it declined rapidly when competing with the high PCN $s_{w90d}$ (Fig 4). Thus, the full-length ancestral R1 does not appear to experience a vertical transmission disadvantage against streamlined variants purely because of plasmid size, but is rapidly displaced by the high-copy number variant $s_{w90d}$.

Next, to evaluate horizontal transmission and its contribution to AMR plasmid displacement, we performed an experiment in which evolved streamlined plasmids were initially present in a different clone than R1 plasmid, both variants at low density in the presence of abundant potential recipients. Populations were then diluted in fresh medium every day for 8 days (Fig 5). In these conditions, overall plasmid dynamics will be driven to a large extent by horizontal transmission, as the large majority of cells are initially plasmid-free. Here we used evolved streamlined plasmids that differ in their PCN *copA\** mutations, including variant $s_{m68c}$ which also carries a *finO* mutation, as well as AMR variants R1 and R1$_{finO}$, as we expect this to impact horizontal transmission.

In control populations without streamlined plasmid variants, R1 invaded the population in 3–4 days, whereas R1$_{finO}$ has already invaded after 1 day. In the presence of any streamlined plasmid, R1 spread was stopped or strongly limited at all time points (Fig 5A top). This includes the $s_{w\_e}$ variant which has an ancestral, low PCN and no *finO* mutation, thus no obvious mutation providing a transmission benefit against R1 apart from the AMR region deletion. In contrast, the presence of most streamlined plasmids had little effect on R1$_{finO}$ spread in the population at early timepoints, with only

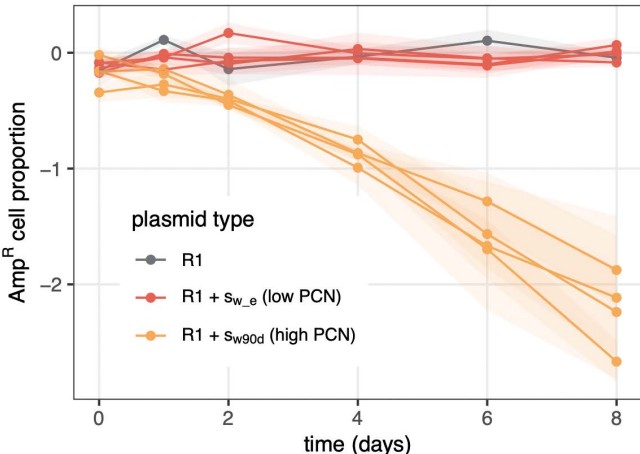

**Fig 4. The outcome of competition at vertical transmission depends on plasmid copy number.** R1-carrying cell proportion of populations seeded with clones containing R1 alone or co-existing with streamlined R1 variants is shown as a function of time. Each line represents an independent founding clone, and shaded areas show standard error from 4 replicate experiments; color indicates initial plasmid content. The data underlying this figure can be found in S1 Data.

$s_{m68c}$ variant limiting AMR densities (Fig 5A bottom). $s_{m68c}$ carries a *finO* mutation similarly to R1$_{finO}$, suggesting that only a streamlined plasmid with transfer derepression 'matching' AMR plasmid transfer derepression can significantly impact early spread. Indeed, focusing on the dynamics of AMR in the recipient (initially plasmid-free) cell population showed the same pattern as for all AMR cells (Fig 5B), demonstrating that the observed dynamics is due to horizontal transmission to the larger recipient population. Interestingly, streamlined plasmid inoculation still influenced R1$_{finO}$ spread but at later time points, with a decrease in R1$_{finO}$ population density overall and in transconjugants, suggesting this was driven by the large cost R1$_{finO}$ imposes on its host rather than by differences in horizontal transmission. Moreover, screening colonies after 3 days for the presence of streamlined plasmids showed that in competition with the slower R1, all streamlined plasmids had already invaded recipients, whereas in competition with R1$_{finO}$, horizontal transmission of streamlined plasmids was more limited (Fig 5C).

Thus, these experiments showed that all streamlined plasmids transmit faster than the full-length R1 AMR plasmid and readily prevent its spread to recipient cells.

## Streamlined plasmids as an efficient barrier to AMR plasmids

The effect of streamlined plasmids observed above implies that acquisition of streamlined plasmids by susceptible cells makes these cells resistant to further plasmid entry. This effect is expected due to entry exclusion functions commonly expressed by plasmids [13] including R1 [24]. To focus on this effect, we directly tested how the presence of R1 variants in recipient cells affects R1 conjugation and spread into recipient populations, and compared their effect with that of canonical defence systems, restriction-modification (RM) systems (Fig 6). We included two streamlined plasmids differing in their copy number: $s_{w\_e}$ (low PCN) and $s_{w90d}$ (high PCN), as well as R1$_{TcR}$, a variant of R1$_{wt}$ with tetracycline resistance replacing chloramphenicol resistance; and two Type II RM systems carried on small cloning, non-conjugative plasmids, that were among the most effective against plasmid conjugation in our previous study [25].

In short-term 1h assays, the presence of R1 variant plasmids (R1$_{TcR}$, $s_{w\_e}$ or $s_{w90d}$) in recipient cells limited conjugation from both R1$_{wt}$ and R1*finO* by approximately 10-fold (Fig 6A), significantly less than the presence of RM systems (EcoRI or EcoRV) (Tukey test on $\log_{10}$ transfer rate ~ plasmid × defence type, RM systems compared to R1 variants,

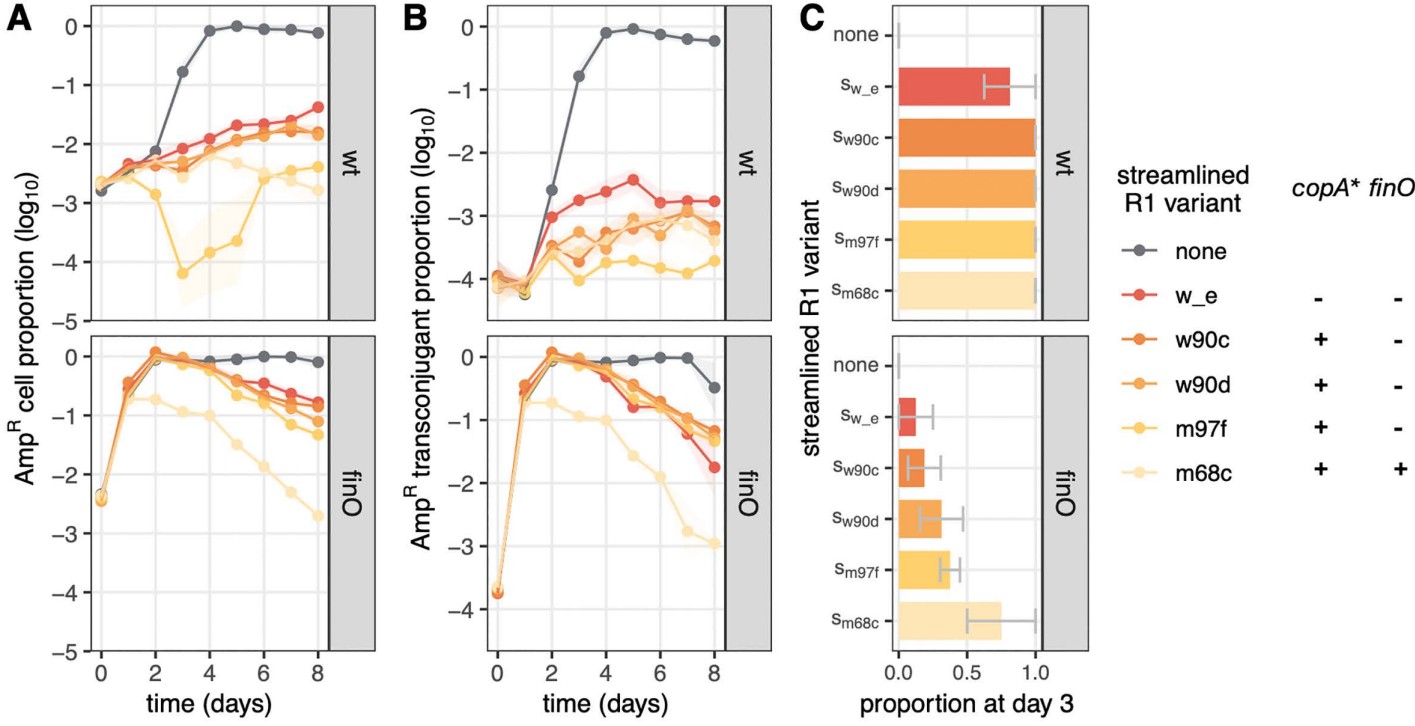

**Fig 5. Streamlined plasmids invade horizontally faster than ancestral plasmids.** Populations were seeded with 1% AMR plasmid-carrying cells, 1% streamlined plasmid-carrying cells (variant indicated by color and detailed in **C**), and 98% plasmid-free cells. Total AMR cell proportion is shown in **A**, and proportion of AMR transconjugants is shown in **B**, with lines and shaded areas indicating, respectively, the average and standard error from 4 replicate experiments. **C** shows the proportion of clones carrying streamlined plasmids at day 3 (4 colonies tested for each of 4 replicate experiments, error bars show standard error across replicate experiments). The data underlying this figure can be found in S1 Data.

difference $= 0.78 \pm 0.26$, $p_{adj} = 0$). However, this pattern was reversed at longer timescales: in a 24 h experiment, transfer towards RM recipients increased over time more than towards R1 variants (Fig 6B), leading to higher transconjugant density at later timepoints (Tukey test on $\log_{10}$ transconjugant density at 24 h ~ plasmid × defence type, RM systems compared to R1 variants, difference $= 1.66 \pm 0.79$, $p_{adj} = 2.3 \times 10^{-5}$). In both assays, the high PCN variant $s_{w90d}$ acted as a stronger barrier than low PCN $R1_{TcR}$ or $s_{w\_e}$ plasmids, suggesting that high PCN of the streamlined plasmids favors their exclusion of AMR plasmids. In contrast, $s_{w\_e}$ and $R1_{TcR}$ displayed very similar efficiency, suggesting that the deletion of AMR genes itself has little impact on defence efficacy, and defence is overall driven simply by exclusion functions carried by all R1 variants. The decline in efficiency of RM systems as a barrier towards AMR plasmids is likely due to the fact that rare early transconjugants will be methylated and free to conjugate further into RM populations. Overall, we conclude that streamlined plasmids are more efficient as a barrier to the AMR plasmids they are evolved from, compared to RM defence systems.

## Both vertical and horizontal transmission advantages can drive plasmid displacement

To understand the potential generality of our experimental results, we built a model of competition between two plasmid variants: a wild-type plasmid (w) and a mutant plasmid (m). Both plasmids replicate within hosts and conjugate to plasmid-free cells; they also conjugate to plasmid-carrying cells at a reduced rate due to entry exclusion. Plasmid m can display a vertical and/or horizontal advantage in transmission compared to plasmid w (Fig 7A). This simple model shows that for any non-null advantage, the m plasmid eventually replaces the w plasmid, and replacement happens faster with

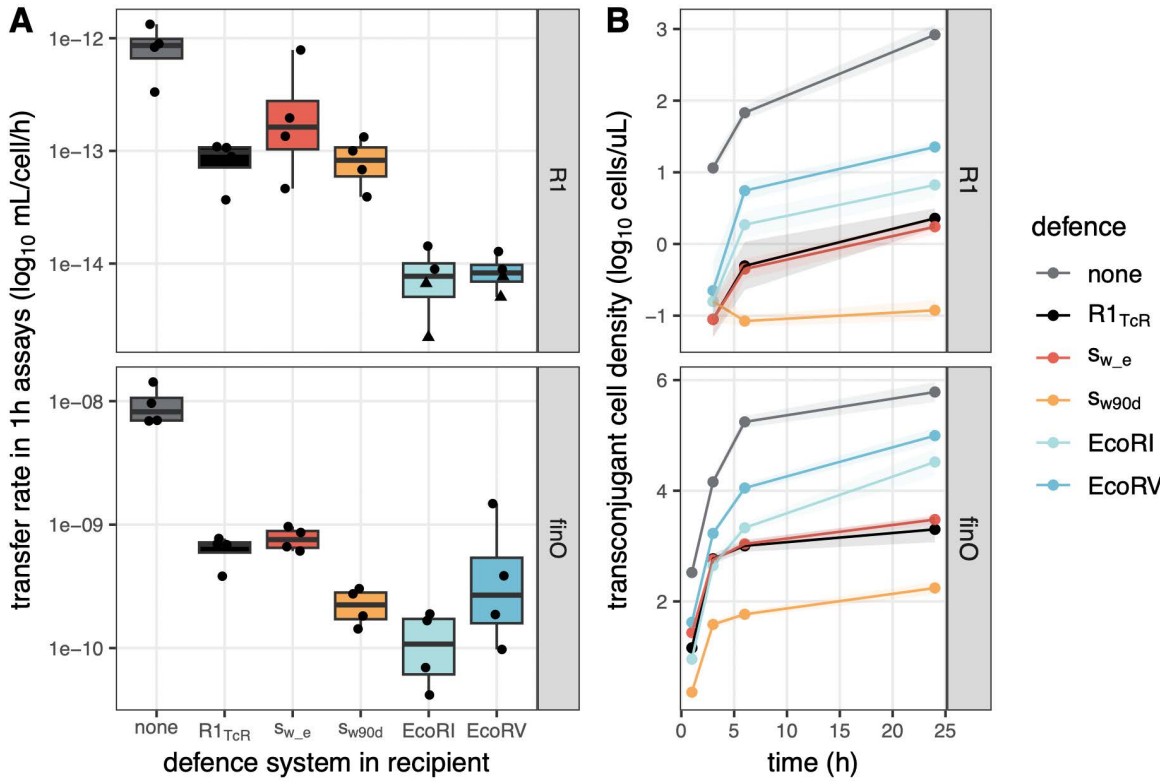

**Fig 6. Streamlined plasmids are a more efficient barrier to AMR plasmids than RM defence systems.** Conjugative transmission of R1$_{wt}$ (top) and R1$_{finO}$ (bottom) was followed towards recipients carrying either no defence, conjugative R1 variants, or RM systems, indicated by color. In **A**, transfer efficiency was measured in 1 h conjugation assays. The center line of the boxplots shows the median, boxes show the first and third quartile, and whiskers represent 1.5 times the interquartile range; individual data points are shown as dots ($n = 4$). In **B**, the density of transconjugant cells was followed over 24 h after seeding populations with 1% AMR plasmid-carrying cells and 99% recipients, with lines and shaded areas indicating, respectively, the average and standard error from 4 replicate experiments. The data underlying this figure can be found in S1 Data.

larger advantages (Fig 7B). The effect of horizontal and vertical advantage depends on plasmid characteristics, with high baseline conjugation rate, higher competition during partitioning and lower entry exclusion leading to faster replacement (S6 Fig). Moreover, it also depends on host population dynamics: modeling an influx of plasmid-free hosts (similar to our experiments in conditions favoring the evolution of high transmission) makes replacement happen faster for any non-null horizontal advantage (S7 Fig).

## Discussion

Previously, we analyzed the ecological and evolutionary dynamics of R1 plasmid exposed to regular immigration of plasmid-free recipients. Using antibiotic resistance as a phenotypic marker for R1 carriage, we observed evolution of high PCN, associated with increased conjugation rate and antibiotic resistance [21]. Yet, R1 population size still declined over time in the face of cell immigration. Here, we show that this decline in resistance was due to rapid evolution of streamlined plasmids with deletions of the AMR region (undetectable by standard assays using resistance to antibiotics to detect plasmid carriage), which displaced ancestral resistance plasmids. Thus, at the population level evolution ultimately led to AMR extinction despite a transient increase in AMR associated to high PCN.

To understand why streamlined plasmids displaced AMR plasmids, we studied the dynamics of R1-carrying cells when exposed to streamlined plasmid variants. At vertical transmission, a high PCN streamlined variant rapidly displaced R1.

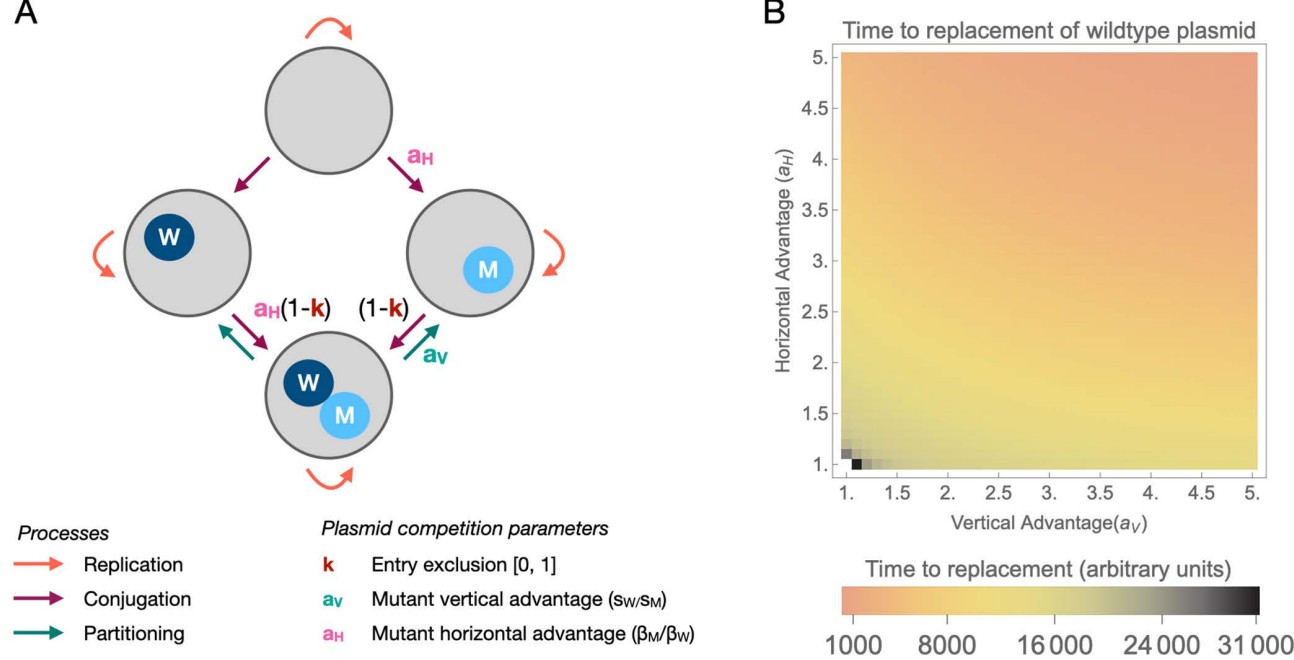

**Fig 7. A model of competition between the wildtype and mutant plasmid. A.** We model cell replication and death, and plasmid conjugation and partitioning. The figure highlights the processes affected by parameters relating to plasmid competition. The $k$ parameter describes the strength of entry exclusion (i.e., 1- susceptibility of cell to co-infection relative to single infection). The mutant plasmid may be associated with an advantage in vertical and/or horizontal transmission. A vertical advantage, $a_V$, reflects a higher probability of being passed on during replication. A horizontal advantage, $a_H$, reflects a higher conjugation rate. **B.** Time (in arbitrary units) taken for the mutant plasmid to replace the wild-type plasmid (defined as the density of the wildtype falling below 0.01 cells per unit volume) as a function of horizontal and vertical advantage. White indicates no replacement. Parameter values: $\rho = 1, c = \frac{1}{10}, \gamma = \frac{1}{10}, K = 1, \beta_W = 1, s_M = 0.1, k = 0.99$, with $\beta_M = a_H \beta_W$ and $s_W = a_V s_M$. We are interested in qualitative insights, the parameters are in arbitrary time units. The data underlying this figure can be found in S2 Data.

This is expected as plasmids are competing not only for replication itself but also for partitioning to daughter cells, and early experiments showed that high PCN mutants of R1 displaced R100, another IncFII plasmid, faster than R1 did [26]. In contrast, R1 seems to be unaffected by the low PCN $s_{w\_e}$ variant. This demonstrates a direct effect of high PCN on vertical transmission; however, our assay has low sensitivity and low PCN plasmids might still have some significant, if lower, role in R1 displacement. More generally, different PCN mutants might interact with the ancestral plasmid differently: for instance, different *copA* mutations can lead to displacement of either the wildtype or mutant plasmid depending on the details of molecular interactions between the *copA* RNA and its target [27,28]. It is likely that only variants able to displace (or at least not be displaced by) the ancestral plasmid will be observed in experimental evolution.

During horizontal transmission, all the streamlined variants we tested—including the low PCN $s_{w\_e}$—spread faster than the ancestor AMR plasmid. A smaller plasmid size might directly be responsible for increased horizontal transmission, possibly because it allows conjugative replication itself to complete faster. Still, we must note that $s_{w\_e}$ also carries another mutation, a G deletion in a polyG tract within the leading region (S1 Table). This mutation was frequent across evolved clones [21] and we cannot exclude that it contributes to $s_{w\_e}$'s horizontal transmission benefit. Ultimately, the effect of plasmid length would need to be tested using engineered plasmids of various lengths. High PCN streamlined plasmids still spread faster than $s_{w\_e}$, in accordance with our previous results showing that high PCN leads to higher conjugation rates [21].

Growth rate assays show that streamlined plasmids have a limited cost, mostly dependent on other mutations present in the evolved plasmids—with *finO* mutations (expected to lead to derepression of transfer) being particularly costly,

especially when combined with high PCN. Comparing a high-copy streamlined variant and a high-copy AMR variant showed a lower cost of the streamlined variant suggesting that at least in high-copy plasmids, AMR gene deletion will bring fitness benefits to the host (and in turn to plasmids themselves via increased vertical transmission). We did not observe any cost for the ancestral R1 plasmid, preventing the detection of any effect of the deletion on plasmid cost in low-copy variants. Yet, deletions of AMR genes can improve plasmid cost in other low-copy plasmids, in particular in the case of highly-expressed β-lactamases [29–31]. Ultimately, the effect of plasmid variants on their hosts could be measured using competitive fitness assays between hosts carrying different plasmid variants. Moreover, whilst our horizontal transmission assay (Fig 5) was designed to maximize the contribution of plasmid conjugation to plasmid competitive success, other factors, including plasmid fitness costs, can also impact their results. Plasmid acquisition cost, a cost experienced shortly after transfer and distinct from long-term fitness cost [32], might play a role. The acquisition cost is negligible in R1 [32], but may be relevant to competitive dynamics between full-length and streamlined plasmids in other plasmid types. Previous studies have shown that both within- and between-host competition contribute to plasmid fitness [33,34], in the absence of horizontal transmission. Here, we show that in the case of conjugative plasmids, variants can compete for vertical as well as horizontal transmission.

Our model shows that any vertical or horizontal transmission benefit is sufficient for a streamlined plasmid to ultimately displace a full-length plasmid, even without assuming any difference in plasmid cost. The vertical transmission advantage displayed by streamlined variant $s_{w90d}$ in competition with co-resident plasmids is dependent on its high PCN and might not be generalizable to other plasmid types which do not easily evolve high PCN, or for which high PCN does not translate into exclusion of a low PCN variant. By contrast, the horizontal transmission advantage was observed for all streamlined plasmids, and might be a property of any conjugative plasmid variant with smaller accessory regions in competition with large, multidrug-resistant plasmids. Model results indicate that the benefits of increased horizontal transmission in streamlined variants will be magnified in conditions that generally promote horizontal transmission itself: high basal rate of transfer, low entry exclusion, or higher influx of plasmid-free immigrant hosts. Moreover, we have studied plasmid competition only in the absence of selection for full-length plasmids. Antibiotic treatment is likely to favor AMR plasmids both by direct selection for AMR plasmid carriage, and by reducing the density of immigrant hosts.

In our evolution experiment, we observed deletions in almost all clones with high PCN, suggesting R1 has a high propensity to experience deletions. Indeed, previous evolution experiments with R1 observed deletions in a similar immigration design [35] or at longer timescales in the absence of immigration [36]. Early molecular studies also showed that R1 and other IncF plasmids can spontaneously dissociate into elements corresponding to its backbone and its AMR region, due to recombination between the IS1 sequences [37,38]. More generally, ISs are generally abundant on plasmids, and specifically enriched in conjugative plasmids encoding AMR genes, with the regions surrounding AMR genes being particularly dense in ISs [39]. Deletions of AMR genes due to recombination between ISs might thus be a general phenomenon across conjugative plasmids, and have been observed experimentally, e.g., [40,41]. Chromosomal rearrangements leading to deletions can also arise from IS recombination [42]. However, plasmids might be particularly prone to deletions, due precisely to their richness in large DNA repeats and presence in cells at high copy number [43]. In our experiments, high PCN likely evolved first - being selected due to its benefits for horizontal transmission [21]—which then promoted recombination and evolution of streamlined plasmids. The emergence of streamlined variants within cells that still contain full-length variants then likely leads to complex eco-evolutionary dynamics with extended periods of co-existence, as observed recently in [34]. It remains to be determined how within-cell and between-cell selection interact, and how long high PCN variants remain stable. Our results are partially similar to the evolution of satellite plasmids observed in [44]. In both cases, shorter plasmids evolved due to rapid recombination and deletion, then competed within cells with their ancestors. However, a key difference is that satellite plasmids in [44] experienced deletions that included replication functions, making them parasites of the larger plasmids as they were incapable to replicate on their own. By contrast,

streamlined plasmids are not only self-sufficient, but display better transmission compared to their ancestors. Accordingly, they evolve and spread in conditions in which horizontal transmission is beneficial.

Overall, streamlined plasmids invade populations more efficiently than their AMR ancestor and when present in a recipient cell, they act as an effective barrier to AMR plasmid conjugation. This barrier effect likely arises mostly from entry exclusion by plasmids present in the recipient cell [13]. In the F plasmid, which transfer machinery is similar to R1's, entry exclusion decreases transfer 100- to 300-fold [45]. This can explain how the initial faster spread of streamlined variants via horizontal transmission translates into lasting limitation of AMR spread. By contrast, plasmids rapidly escape restriction by two RM systems. This is consistent with our previous results, in which median plasmid restriction during short-term conjugation assays across plasmids and RM systems in *E. coli* was only 14-fold [25]. RM systems were carried on small cloning plasmids, thus might be expressed at higher levels than in natural isolates, although these RM systems were initially found on small, mobilizable medium copy number plasmids [46]. Plasmids have evolved multiple anti-defence mechanisms [47] as well as defence avoidance strategies [48]. Avoiding competition arising from core replication mechanisms or from entry exclusion encoded by the plasmid immediate ancestor is likely harder to achieve. Moreover, host defences are likely under little selection pressure to evolve to target plasmids (much less costly to host cells than phages), whereas plasmids have evolved to compete with each other for host resources [49]. AMR and streamlined plasmids are effectively competing for the host niche, translating into exclusion of AMR plasmids by streamlined ones, similar at a lower selection level to the replacement of AMR strains by competing non-AMR ones in fecal transplants [50].

Our results suggest that plasmid competition might serve to limit the spread of mobile AMR, whether in the context of natural populations, or through engineering approaches. AMR plasmids can cohabit naturally with non-AMR, highly related plasmids within populations [51], providing opportunities for competitive exclusion dynamics. For instance, in a genomic study of bloodstream isolates, an antibiotic-susceptible lineage was found to contain a non-AMR IncF plasmid related to the AMR plasmids carried by other, resistant isolates, suggesting that the susceptible plasmid excluded AMR plasmids [52]. In a CRISPR-based engineering approach, supplying a mobilizable plasmid incompatible with the target plasmid also helped prevent the spread of AMR, especially in conditions of horizontal transmission [53]. Streamlined plasmids evolving from AMR plasmids would have the added benefit of being antibiotic-sensitive, so do not risk disseminating other marker genes into the environment. Finally, at longer timescales evolution of streamlined plasmids and their displacement of competing plasmids carrying various accessory genes could contribute to the pattern of accessory gene depletion recently observed in plasmids compared to chromosomes [54] and shape the broader mobile gene pool.

## Materials and methods

### Strains, plasmids, and growth conditions

The ancestral strains for our repeat evolution experiment were the same as in our first evolution experiment [21]: *E. coli* MG1655 (wt) and MG1655 *mutL*::KnR (mut) as initial R1-carrying strains, and wt and mut variants marked with *td-Cherry* as plasmid-free recipients. The mut strain is a mutator with approximately 100-fold elevated mutation rate [55], and was initially chosen in [21] to allow for faster evolution; we indeed observed more mutations in mut evolved clones, but no qualitatively different patterns [21]. For shorter-term competition experiments, several host strains were used as detailed below: MG1655, MG-*tdCherry* and MG1655 Δ*lac* as described in [21] and a spontaneous rifampicin-resistant mutant of MG1655 (MG1655 Rif$^R$ [10]) were also used. MG1655Δ*dapA*::ErmR was used as donor for conjugation experiments, and MG1655 Δ*recA*::frt for some colony PCR assays. To test the effect of RM systems on transmission, pEcoRI and pEcoRV plasmids (derived from pBR322), as used in [25], were transformed into MG1655 Δ*lac*. As AMR plasmids for transmission experiments, we used R1 (ancestor of the evolution experiments) as well as the R1$_{finO}$ mutant described in [21]. R1$_{TcR}$ was obtained by λred recombination [56]: the region of RP4 plasmid containing the *tetR* and *tetA* genes was amplified by PCR (using the region between coordinates 12976 to 15507 of reference sequence BN000925) and used to replace

the *cat* gene in R1 (coordinates 39757 to 40484 in R1 reference sequence KY749247). We also used evolved plasmids sequenced from our first evolution experiment, after transfer by conjugation to other host backgrounds. Nomenclature for evolved clones follows [21], using *bNx*, where *b* = host background (w for wt, m for mut), *N* = evolution treatment (99, 98, 90, 68 or 0 as % of plasmid-free hosts added at each passage, _ for the no plasmid control treatment) and *x* = lineage (a to f). Streamlined plasmid variants with deletion of the full AMR region are indicated as s. Details of evolved R1 variants used are shown in S1 Table.

Cells were grown in LB medium and clones bearing antibiotic-resistant plasmids were selected with ampicillin 100 mg/L. In the experiments shown in Fig 6, ampicillin was only added to pEcoRI- and pEcoRV-carrying overnight cultures, not in the mixed cultures. Rifampicin was used at 100 mg/L for selection of MG1655 Rif$^R$. 300 µM diaminopimelic acid (DAP) was added to Δ*dapA* cultures. MG-*td-Cherry* colonies were identified by their red fluorescence, and MG1655 Δ*lac* colonies by plating on LB-agar supplemented with IPTG 1mM and X-Gal 0.2g/L. Evolution and competition experiments were done at 37°C, in liquid cultures with shaking at 180 r.p.m., and with 100-fold daily total dilution. In the repeat evolution experiment, for the immigration treatment, plasmid-free immigrants (wt and mut *td-Cherry* strains) were grown fresh from glycerol stock and mixed with the evolving resident cultures in 95:1 ratio at each passage.

### Detection and characterization of plasmid variants

Two approaches were used to detect R1 plasmid variants by colony PCR using DreamTaq PCR mastermix (Thermo Scientific) with primer concentration 0.4 µM. First, primer pairs specific to the backbone of the plasmid (CopA-F & CopA-R, a product indicates that any variant of R1 is present) or to the *bla* gene located in the middle of the AMR region (bla-F & bla-R, a product indicates that the *bla* gene is present) were used for PCR with cycling parameters: 10 min at 95°C, 35 × (15 s at 95°C, 15 s at 52°C, 10 s at 72°C). Second, the R1$_{del}$-F & R1$_{del}$-R primers were used for PCR with cycling parameters: 10 min at 95°C, 30 × (15 s at 95°C, 15 s at 64°C, 20 s at 72°C). On the R1 plasmid, these primers are located more than 36 kb apart, thus expected to yield no product, but they yield a short product when variants with the AMR region deleted are present (S2A Fig). With the PCR settings used, R1$_{del}$ primers yielded a faint band on R1 plasmid, clearly fainter than the band yielded with streamlined plasmids. Sequencing showed that this band corresponded to the same sequence product encompassing the AMR region deletion. To investigate its origin, PCR reactions were run using three additional primer pairs of varying distance to the deleted region (R1$_{del}$_short, R1$_{del}$_mid and R1$_{del}$_long), and on colonies of fresh R1 transconjugants into MG1655 and MG1655 Δ*recA*::frt, using varying PCR annealing temperatures and elongation times, as detailed in S2 Fig legend. The faint band was still present when using different primer pairs (S2B Fig). This suggested that the variant might occur spontaneously at low frequencies in R1 colonies. However, it was also observed using a Δ*recA* host unable to undergo homologous recombination (S2C Fig), and disappeared using one of the primer pairs with more stringent annealing conditions (S2D Fig). This suggested rather that the presence of this faint band is due to some rare recombination events happening *in vitro* and amplified by PCR. Still, the difference in band intensity using R1$_{del}$ primers allowed us to reliably differentiate which colonies carry streamlined plasmids (S2A Fig). For screening PCRs, a negative control (using a R1-carrying colony) and a positive control (using a s$_{w\_e}$-carrying colony) were added to each gel for comparison and accurate determination of s plasmid carriage. For Sanger sequencing (Eurofins), PCR products were purified with Qiagen QIA Quick. Primer sequences are shown in S2 Table.

### Bioinformatic analyses

We reanalyzed Illumina sequencing data on evolved clones from [21]. Unique read coverage from mapping reads to R1 plasmid sequence, generated with breseq as described in [21], was averaged over R1 sequence coordinates 1-5570 and 41387-99378 to obtain coverage of R1 backbone; and over R1 sequence coordinates 6200-15335 and 15950-40664 to obtain coverage of R1 AMR region. *copA*∗ alleles were also manually inspected in Geneious after mapping reads to R1 plasmid sequence, in order to identify rare reads with ancestral sequence.

## Obtaining bacterial clones carrying streamlined plasmids alone

Overnight cultures of evolved clones carrying both full-length and streamlined plasmids (based on Illumina sequencing read depth from [21]) were mixed with the target recipient strain in a 1:25 donor to recipient ratio during 3 h in 500 µL LB broth, then plated without antibiotic selection for AMR plasmid carriage. Recipient background was identified depending on recipient phenotype (red color, white color in presence of IPTG + XGal and rifampicin resistance respectively for *td-cherry*, Δ*lac* and Rif$^R$ strains); and colonies containing only streamlined plasmids were screened for plasmid presence by PCR (presence of a product with R1_CopA primer pair, absence of a product with R1_bla primer pair) and by checking for ampicillin sensitivity.

## Plasmid stability and vertical transmission experiments

To test for plasmid stability, MG Rif$^R$ strains carrying R1$_{wt}$ or streamlined variants s$_{w\_e}$ or s$_{w90d}$ were diluted 100-fold for 8 days from an initial overnight culture, in 1mL LB broth in 24-well plates without antibiotics, in 4 replicates. 8-day populations were streaked out on LB-agar, and plasmid presence was checked by colony PCR on 8 colonies per population, using DreamTaq PCR mastermix and primers parM-F and parM-R at concentration 0.4 µM with cycling parameters: 10 min at 95°C, 35 × (15 s at 95°C, 15 s at 52°C, 20 s at 72°C).

For within-cell competitions between variants with vertical transmission, we first conjugated a streamlined variant into MG1655 Rif$^R$ and screened for MG1655 Rif$^R$ colonies containing the streamlined variant and not containing the full-length (ampicillin-resistant) plasmid. Next, the full-length R1 plasmid (ancestor) was conjugated into this new recipient using ampicillin + rifampicin to select for transconjugants, selecting 4 independent colonies. This was done with evolved clones w_e and w90d. Competitions were run in 200 µL LB broth in 96-well plates. Clones were first inoculated into LB containing 50 mg/L kanamycin (Kn) overnight, then cultures were diluted 100-fold every 24 h into LB without any antibiotic and density of R1-carrying cells was measured using ampicillin resistance as a marker.

## Plasmid horizontal transmission assays

For competitions between variants in conditions favoring horizontal transmission, 200 µL LB broth populations in 96-well plates were seeded with 1% (vol) overnight cultures of MG1655 containing either R1 or R1$_{finO}$ plasmid (m32e_t$_{12}$, an evolved variant with a *finO* mutation only, see S1 Table), 1% overnight cultures of MG Δ*lac* containing either no plasmid or one of s$_{w\_e}$, s$_{w90c}$, s$_{w90d}$, s$_{m68c}$ or s$_{m97f}$, and 98% overnight cultures of MG *td-Cherry*. Cultures were then diluted 100-fold every 24 h into LB without any antibiotic and density of R1-carrying or R1$_{finO}$-carrying cells was measured using ampicillin resistance as a marker. To evaluate the spread of streamlined plasmids at 3 days, colony PCR was performed on colonies grown on LB-agar without antibiotics, using the R1$_{del}$ primer pair.

The effect of R1 variant plasmids and RM systems as barriers to AMR plasmid invasion was measured in two different assays, both using MG1655 Δ*dapA*::ErmR as a donor and MG Δ*lac* as a recipient, in a total volume of 1mL of LB broth supplemented with DAP 120 µM in 24-well plates, and incubated at 37°C with shaking at 180 rpm. Donor, recipient, and transconjugant densities were quantified by plating on LB-agar supplemented with DAP + Chl, no supplement, and Chl, respectively. 1 h conjugation assays were run using 50 µL of donor culture and 50 µL of recipient culture added to pre-warmed medium. Transfer efficiency was estimated as $\gamma$ (mL/cell/h) = $T/DRt$, where $T$, $D$, and $R$, respectively, indicate the cell density of transconjugants, donors, and recipients (cells.mL$^{-1}$), and $t$ is the incubation time (h). For 24 h time course experiments, populations were seeded with 1% (vol) overnight cultures of donors and 99% overnight cultures of recipients. When no transconjugant colony was present, a threshold transconjugant density was calculated by assuming that 0.5 transconjugant colony was detected.

## Effect of plasmids on host growth rate

Strains MG1655 Rif$^R$ and MGΔ*lac* alone or carrying a single plasmid genotype were grown overnight from glycerol stock without antibiotic selection. Each strain was grown independently from glycerol stock in three separate overnight cultures. Cultures

were then diluted 10,000-fold into 200 μL LB in 96-well microplates and covered with 50 μL mineral oil. Optical density at 600 nm was measured at 5-min intervals in a Multiskan SkyHigh plate reader at 37°C for 24 h with shaking on. Maximal growth rate was computed using the R package growthrates [57] between 2 and 10 h post-inoculation, with the h parameter set at 8. Each plasmid-carrying clone was run along its plasmid-free control in the same strain, with 12 technical replicates per growth plate.

Data and statistical analysis used R version 4.3.2 [58]

## Model of plasmid competition

We model the density of cells (i.e., cells per unit volume) carrying no plasmid ($N_0$), carrying the wild-type plasmid ($N_W$), the mutant plasmid ($N_M$), or both ($N_{WM}$). Cells replicate at maximum rate ρ with carrying capacity $K$, and die at rate γ. Plasmid carriage is associated with a fitness cost $c$, which we assume affects replication.

Plasmid conjugation is density-dependent and occurs at rate $β_W$ for the wild-type plasmid and $β_M$ for the mutant plasmid. The mutant plasmid may have an advantage in horizontal transmission, parametrized by $a_H = β_M/β_W$ (with $a_H ≥ 1$). Conjugation to already infected cells results in co-infection. The $k$ parameter describes the strength of entry exclusion (i.e., 1- susceptibility of cells to co-infection relative to single infection). If a plasmid transmits to an already co-infected cell, the incoming variant displaces the other variant with probability 1/2 (this is necessary to ensure structural neutrality [59]). We assume no simultaneous transmission of both plasmids from co-infected cells. Co-infected cells have the same overall conjugation rate as singly infected cells, transmitting each plasmid with probability 1/2.

We assume no total loss of plasmid carriage, but co-infected cells may lose all copies of one plasmid variant during replication, reverting to carriage of a single plasmid. This occurs with probability $s_W$ for the wild-type and $s_M$ for the mutant plasmid (with $s_W + s_M ≤ 1$). The mutant plasmid may have an advantage in vertical transmission, parametrized by $a_V = s_W/s_M$ (with $a_V ≥ 1$).

In the supplement, we also explore the impact of an influx of plasmid-free cells at rate ε. The influx is 0 in all simulations except S7 Fig.

The model is described by the following equations:

$$dN_0/dt = N_0 \left[ ρ \left( 1 - T/K \right) - γ - λ_M - λ_W \right] + \epsilon$$

$$dN_W/dt = N_W \left[ (1-c)ρ \left( 1 - T/K \right) - γ \right] + N_0 λ_W - (1-k)N_W λ_M + N_{WM} s_M (1-c)ρ \left( 1 - T/K \right)$$

$$dN_M/dt = N_M \left[ (1-c)ρ \left( 1 - T/K \right) - γ \right] + N_0 λ_M - (1-k)N_M λ_W + N_{WM} s_W (1-c)ρ \left( 1 - T/K \right)$$

$$dN_{WM}/dt = N_{WM} \left[ (1 - s_W - s_M) (1-c)ρ \left( 1 - T/K \right) - γ \right] + (1-k) \left[ N_W λ_M + N_M λ_W \right]$$

where $λ_i = β_i \left( N_i + N_{WM}/2 \right)$ and $s_W + s_M ≤ 1$.

All simulations were started at the equilibrium density of the wild-type plasmid in absence of the mutant plasmid ($N_0 = 0$ cells per unit volume and $N_W = 0.89$ cells per unit volume for the standard parameters used in the main text) and introduction of a small quantity of coinfected cells ($N_{WM} = 0.01$ cells per unit volume). Model simulations were run in Mathematica 14.2 [60]. The code is available as a (S2 Data).

## Supporting information

**S1 Table. R1 variant plasmids used in this study.** Mutations present in R1 variants are indicated.
(XLSX)

**S2 Table. Primers used in this study.**
(XLSX)

**S3 Table. Statistical analysis of growth rate data** Results of TukeyHSD tests are shown for all plasmid type comparisons, using a linear model, lm(exponential growth rate ∼ strain × plasmid × rep_nb). Strains are MG Δ*lac* and MG Rif$^R$; rep_nb was coded 1–6 as a factor.
(XLSX)

**S1 Fig. Detail of Illumina sequencing read coverage across R1 plasmid sequence.** Relative coverage of sequencing reads is shown for all clones across R1$_{wt}$ sequence map. Relative coverage was measured as the sum of coverage of both unique and repeat reads, divided by the overall average coverage of reads mapped to the chromosome. The data underlying this figure can be found in S1 Data.
(PDF)

**S2 Fig. Results of colony PCRs designed to detect deletions within R1.** "-" indicates the negative control without bacteria; other reactions were run with *Escherichia coli* carrying no R1 or R1 variants as indicated. Numbers in A indicate the size of some DNA ladder bands, in bp, the ladder used was the same for all gels. Titles in italics indicate primer pairs used in each section. In A and D, annealing temperature was 64°C; in B and C annealing temperature was 56°C. In A and C, extension time was 15 s; in B and D it was 1 min to ensure full amplification using R1_del_long primer pair. In A to C, 30 cycles were performed; in D 35 cycles were performed to enhance detection of any possible faint band.
(PDF)

**S3 Fig. Characterization of coverage variation along sequenced evolved R1 clones.** In **A**, short-read coverage of the AMR region is shown on the y-axis and coverage of the rest of the plasmid is shown on the x-axis; each dot represents a sequenced clone with evolution treatment indicated by color and strain background by dot type (circles = wt, triangles = mut). **B** shows the relative coverage of the AMR region (compared to the backbone region), as a function of the frequency of *copA** mutant allele among sequenced reads for each sequenced clone. Interpretation of plasmid content is shown for three regions of both graphs, with bold lines showing plasmid regions present, thin lines deleted regions, and the green circle indicating the ampicillin resistance marker. The data underlying this figure can be found in S1 Data.
(PDF)

**S4 Fig. Stability of R1 variants after 8 days of passage** Plasmid presence was checked by colony PCR using parM primers on 8 colonies per evolved population. Each gel shows a population replicate number, with plasmids R1, s$_{w\_e}$ and s$_{w90d}$; rep b gel also includes a no DNA control, a plasmid-free bacteria control and a R1-carrying control, as indicated.
(PDF)

**S5 Fig. Detail of growth curves data and exponential growth rate across replicates.** In **A**, growth curve data are shown for each plasmid in color. The portion of the curves used for calculating exponential growth rate is shown in light gray. In **B**, calculated exponential growth rates are shown for all strains. Each data point corresponds to an individual technical replicate (well within a 96-well plate); the center line of the boxplots shows the median, boxes show the first and third quartile, and whiskers represent 1.5 times the interquartile range, calculated across $N = 12$ technical replicates, with different colors indicating independent biological replicates using different overnight cultures. The data underlying this figure can be found in S1 Data.
(PDF)

**S6 Fig. The impact of plasmid characteristics on competition between the mutant and wildtype plasmids.** The heatmaps shows the time (in arbitrary units, log10 transformed) taken for the mutant plasmid to replace the wildtype plasmid (defined as the frequency of the wildtype falling below 1%) as a function of horizontal and vertical advantage.

Each row shows the impact of setting one parameter differently from the standard values used in the main text. Top: transmission rate of the wildtype plasmid ($\beta_W$). Middle: partitioning loss of the mutant plasmid ($s_M$). Bottom: Entry exclusion ($k$). We also show the log10 transformed main text results (standard parameters) for reference. Standard parameter: $\rho = 1$, $c = \frac{1}{10}$, $\gamma = \frac{1}{10}$, $K = 1$, $\beta_W = 1$, $s_M = 0.1$, $k = 0.99$, with $\beta_M = a_H \beta_W$ and $s_W = a_V s_M$. We are interested in qualitative insights; the parameters are in arbitrary time units. White indicates no replacement. We have used a log scale to fully capture the extent of variation. The data underlying this figure can be found in S2 Data. (TIFF)

**S7 Fig. The impact of an influx of plasmid-free cells on competition between the mutant and wildtype plasmids. A**. Schematic of the model, with the addition of an influx of plasmid-free cells (green arrow) in units of cells/(volume * time). The heatmaps shows the time (in arbitrary units) taken for the mutant plasmid to replace the wildtype plasmid (defined as the density of the wildtype falling below 0.01 cells/unit volume) as a function of horizontal and rate of influx of plasmid-free cells. Parameters values: $\rho = 1$, $c = \frac{1}{10}$, $\gamma = \frac{1}{10}$, $K = 1$, $\beta_W = 1$, $s_M = 0.1$, $s_W = 0.1$, $k = 0.99$, with $\beta_M = a_H \beta_W$. We are interested in qualitative insights, the parameters are in arbitrary time units. White indicates no replacement. The data underlying this figure can be found in S2 Data. (TIFF)

**S1 Data. Supporting data associated with experimental data.** (XLSX)

**S2 Data. Mathematica notebook associated with modeling data.** (NB)

**S1 Raw Images.** Raw gel images associated with S2 and S4 Figs. (PDF)

# Acknowledgments

We thank Chantal Lotton for the MG1655 Δ*recA*::frt strain.

# Author contributions

**Conceptualization:** Andrew C. Matthews, Tatiana Dimitriu.

**Data curation:** Andrew C. Matthews, Tatiana Dimitriu.

**Formal analysis:** Andrew C. Matthews, Tatiana Dimitriu.

**Funding acquisition:** Tatiana Dimitriu.

**Investigation:** Andrew C. Matthews, Sonja Lehtinen, Tatiana Dimitriu.

**Methodology:** Andrew C. Matthews, Sonja Lehtinen, Tatiana Dimitriu.

**Project administration:** Tatiana Dimitriu.

**Visualization:** Tatiana Dimitriu.

**Writing – original draft:** Sonja Lehtinen, Tatiana Dimitriu.

**Writing – review & editing:** Andrew C. Matthews, Sonja Lehtinen, Tatiana Dimitriu.

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
