## [Editor Report · Decision Letter 0]

30 Apr 2025

Dear Dr Dimitriu,

Thank you for submitting your manuscript entitled "Stealth plasmids: rapid evolution of deleted plasmids can displace antibiotic resistance plasmids under selection for horizontal transmission" for consideration as a Research Article by PLOS Biology. Please accept my apologies for the delay in getting back to you with feedback as we consulted with an academic editor about your submission.

Your manuscript has now been evaluated by the PLOS Biology editorial staff, as well as by an academic editor with relevant expertise, and I am writing to let you know that we would like to send your submission out for external peer review.

Once your full submission is complete, your paper will undergo a series of checks in preparation for peer review. After your manuscript has passed the checks it will be sent out for review. To provide the metadata for your submission, please Login to Editorial Manager (https://www.editorialmanager.com/pbiology) within two working days, i.e. by May 02 2025 11:59PM.

Kind regards,

Richard

Richard Hodge, PhD

rhodge@plos.org

PLOS

---

## [Decision Letter · Decision Letter 1]

18 Jun 2025

Dear Dr Dimitriu,

Thank you for your patience while your manuscript "Stealth plasmids: rapid evolution of deleted plasmids can displace antibiotic resistance plasmids under selection for horizontal transmission." was peer-reviewed at PLOS Biology. Please accept my sincere apologies for the delays you experienced during the peer review process. Your manuscript has now been evaluated by the PLOS Biology editors, an Academic Editor with relevant expertise, and by three independent reviewers.

In light of the reviews, which you will find at the end of this email, we would like to invite you to revise the work to thoroughly address the reviewers' reports.

As you can see, the reviewers think the work is interesting and well done, but they each raise important and overlapping concerns that should be addressed in the revision. Specifically, Reviewer #1 notes that additional data to support the generality of the importance of the plasmids in counteracting AMR is needed, including additional modelling and analyses of plasmid databases. In addition, the reviewers raise overlapping concerns with the accuracy and use of the term ‘stealth plasmid’ and note conceptual similarities with the findings of Zhang et al (PMID 31863068). We ask that the distinction between the current study and Zhang et al is discussed and clarified in the revised version. Finally, they raise concerns with the overall presentation of the manuscript and provide suggestions to improve the accessibility/readability of the article.

Given the extent of revision needed, we cannot make a decision about publication until we have seen the revised manuscript and your response to the reviewers' comments. Your revised manuscript is likely to be sent for further evaluation by all or a subset of the reviewers.

**IMPORTANT - SUBMITTING YOUR REVISION**

*Re-submission Checklist*

*Published Peer Review*

*PLOS Data Policy*

*Blot and Gel Data Policy*

Best regards,

Richard

Richard Hodge, PhD

rhodge@plos.org

REVIEWS:

Reviewer #1: In this manuscript, Dimitriu and Matthews re-analyse samples from a previous experiment to investigate the evolution of plasmid-borne antimicrobial resistance. In that previous work, they identified that plasmid-borne antimicrobial resistance required the evolution of increased horizontal transmission and copy number in order to persist. In the current work, they unpick some of the underlying mechanism, and show the emergence of 'stealth' plasmids, lacking resistance genes, which could effectively outcompete resistance-encoding plasmids. This is an intriguing finding that meaningfully extends the previous work. The paper was clearly and engagingly written and presented. However, I found that several aspects of the current work could be better explained with reference to the previous study (oftentimes, elements of the current work seemed somewhat arbitrary to readers not familiar with the previous paper). I also think some further analyses could help generalise some of the findings further.

Main comments:

- The main finding is that 'stealth plasmids', which do not confer AMR but are able to outcompete AMR plasmids in cells and in populations, emerge quickly by deletion and could help to block the transmission of AMR plasmids. This is an interesting finding, but it is hard to see how generalisable it is, i.e. to different plasmids or to natural populations. A couple of further analyses would extend the impact of the current work for the broader audiences of this journal. For example:

 I wonder if it would be possible to adapt one of the well-developed mathematical models of plasmid dynamics to assess the conditions under which a stealth plasmid can prevent the maintenance of an AMR plasmid — I wonder if such a dynamic could be predicted, based on the extent to which the stealth plasmid impacts horizontal and vertical fitness?

 What is the evidence for 'stealth plasmids' (i.e. cryptic plasmids with very high identity to sympatric AMR-encoding plasmids) in plasmid databases or in metagenomes?

I appreciate that these each would constitute a significant amount of work. However, though readers like myself would certainly be interested in the work in its current form, without some broader context, I am not sure whether the findings are sufficiently general for the wide readership of PLOS Biology.

- The finding has some conceptual resemblances to Zhang et al. (2019, doi: 10.1038/s41467-019-13709-x), which should be discussed.

Other comments/suggestions:

- It would be useful to give some wider context to R1. What kind of plasmid is it, are any similar plasmids involved in clinically important AMR? Etc.

- Line 73ff could be rewritten to better emphasise novelty/rationale of the current study.

- Line 75: define 'stealth plasmid', or put in inverted quotes. Is a 'stealth plasmid' the same as a cryptic plasmid? What is the difference with the 'satellite plasmids' of Zhang et al. 2019?

- Line 88 ff. is in present tense. It is maybe a personal preference, but I would rather see all the results section in the past tense, describing specific work that has been done.

- Line 100. "Deleted plasmids" could imply the whole plasmid was deleted. Maybe "plasmid deletions"?

- Line 100ff describes contamination — how reliable is the rest of the experiment, in light of this observation? This finding could be explained a bit further for a wider audience, who may not understand the meaning of "nearby" (presumably this is an adjacent well in the 96-well plate?)

- Line 109. I feel more could be done to get to the bottom of the faint band observed in the ancestral R1 plasmid. Does this occur in colony PCR? Does it occur with other primer designs? Is it possible to estimate the rate of excision, e.g. through use of qPCR?

- Fig. 2. What does 'mut' refer to? I understand this is discussed in the methods section and in the previous paper. I think a bit more could be done to make the current work stand more independently, by briefly explaining the justification for this design.

- Line 210. Typo: "extent"

- Line 212. Which stealth mutant is a finO derepressed mutant? This could be indicated here, when the concept is first introduced.

- Line 213. "Expect this to impact horizontal transmission" — what exactly? The copA mutations or finO? Clarify.

- Line 248. The rationale for testing with ∆lac and RifR is not made clear. Which was used in the evolution experiments? Again, a few words could explain. Why are there gaps, i.e. why were some plasmids only tested in some hosts, and not both? Perhaps it would be better to reorient the subplots in Fig. 5 so they are placed horizontally, which would allow you to drop values from the x-axes that do not apply (e.g. sm97f/RifR), and enable better comparisons on the y-axis across all strains.

- Line 252. Formatting of p-values needs correction (there is a 'times' symbol missing).

- Fig 5 is not consistent with text. Figure shows sw90d and sm97f as orange points with red bars, which according to the legend describes plasmids that have repressed transfer and high PCN. However, in the text these plasmids are referred to as "low PCN stealth variants" (line 265).

- Section beginning line 246 concludes that cost to the host is determined by transfer gene expression. This finding is drawn from the observation that plasmids with derepressed transfer (finO, sm68c) are more costly. However, we don't actually know what the transfer rate (or indeed transfer gene expression) is in these clones, only that they invade populations faster than the ancestral plasmids.

- What threshold is used to define 'high PCN' and 'low PCN', and why?

- Fig 6 was a little confusing — it was hard to see the points for the stealth plasmids in the left hand sub panel of panel B. Perhaps if you include the limit of detection on the y-axis , then these points could be shown more visibly with the connecting lines.

- Section beginning line 287 is interesting and I like how you contextualise the findings in a broader context of MGE competition. As discussed above, it would be exciting to generalise this further e.g. using a mathematical model, and/or analysis of published sequences.

Reviewer #2: The manuscript 'Stealth Plasmids: Rapid Evolution of Deleted Plasmids Can Displace Antibiotic Resistance Plasmids Under Selection for Horizontal Transmission' by Matthews & Dimitriu is both interesting and highly relevant for researchers working in bacteriology, plasmid biology, antibiotic resistance, and molecular evolution. I have several major and minor comments that I hope the authors can use to improve the manuscript.

Comments/Concerns:

1. Readability:

a. While I understand the authors' desire to reference and discuss their previous work (PNAS 2021), this section makes the manuscript slightly tedious to read in the beginning. Could the authors consider limiting the discussion of the 2021 paper to a few key points and moving the detailed discussion to the Discussion section?

b. Regarding the above, is the lower part of Figure 1A necessary?

c. Line 196: The manuscript refers to Fig. 3B, but this figure does not appear.

d. I do not think the term "Stealth Plasmids" is appropriate for the modified plasmids. While these plasmids may have appeared 'stealth' from the perspective of the PNAS 2021 paper authors, the term does not convey the correct meaning for readers of this manuscript. It implies that the plasmids lose antimicrobial resistance (AMR) as a strategy, which is not the case.

e. I also disagree with the use of the term "deleted plasmids." To me, this implies that the plasmids are completely absent from the cell, when in fact they are present but have portions of their sequence deleted.

f. The names of the strains were difficult to follow and understand until the Methods section (which appears after the Discussion only). In the Methods section, these names are well explained (lines 426-428). I suggest that the strain names be introduced earlier, preferably in the Results section.

g. While reading the Results section, I assumed that "mut" referred to "mutator." This is only clarified in the Methods section.

h. For Figures 2B and 4B, the horizontal axes appear to represent frequency (ranging from 0 to 1), not percentages (0 to 100), correct?

i. I think Fig. 5 extremely difficult to read. Moreover, I think the legend is wrong at "low PCN and high PCN": I think dots should be there, not boxes.

j. The vertical axis of Fig 5 is h^-1, not h-1. On the right of Fig.5 dlac should delta-lac.

k. Line 268: missing "x" in p=6.8x10-6 and in line 271 p=3.7x10-5. The same in line 279 and possibly other places

L. line 293 "include 3…" should be "include three…". The same in line 308

m. 1st paragraph of Discussion needs the reference to PNAS 2021 paper.

2. In the Results section, it was unclear where the RM genes were located. I initially thought they might be on the R1 plasmid (or in the mutant plasmids), or in the chromosome. However, it turns out they are on artificial non-conjugative plasmids (pEcorRI and pEcoRV, as explained in the Methods section). I think this should be briefly mentioned in the Results section. Could the authors discuss how the location of these RM systems on plasmids might influence the results?

3. The pEcorRI and pEcoRV plasmids were "maintained using their ampicillin resistance marker" (line 423). When exactly was the ampicillin used? Was it just before the experiments in Fig. 5, or was it used during the 3 days mentioned in Fig. 5? This could potentially affect the results.

4. I am concerned that in several experiments, the authors concluded that the AMR region was absent simply because plasmid-bearing cells were ampicillin-sensitive. However, the plasmids could still carry resistance genes for the other antibiotics typically carried by the R1 plasmid (e.g., sulfonamides, streptomycin, spectinomycin, kanamycin, chloramphenicol, and, of course, ampicillin). Figure 1C suggests that losing ampicillin resistance (AmpR) implies losing all resistance markers, but this is not certain, is it?

Reviewer #3: The authors of this study follow up on the results found in a previous study, which found that antimicrobial resistance encoded on conjugal plasmids was lost from the population under regular recipient invasion (Dimitriu et al., 2021). In their recent study the authors add another condition to their former results and investigate the part of the evolved population, that did retain the plasmid backbone, but lost the antimicrobial resistance genes due to a recombination event between two ISs on R1. Their experiments suggest an advantage in vertical and horizontal transfer due to multiple factors like increased plasmid copy number by comparing several stealth plasmid variants and the ancestral R1. Furthermore, they show that stealth plasmids do provide a more efficient barrier against incoming related plasmids compared to restriction-modification-systems.

Major comment

I find the term 'stealth plasmid' inaccurate - that R1 plasmid variant is lacking an AMR island, but is still able to replicate and conjugate (not as e.g., described in Zhang et al. 2019 Nat Commun). So, in essence, the authors are following here the dynamics of several R1 variants.

The specific experiments are well designed and rather easy to follow. However, as mentioned below, quite few controls seem to be missing, which makes it hard to evaluate its true relevance i.e. evolution of stealth plasmids. Nevertheless, the authors give an example of the complexity of factors like plasmid copy number and gene expression influence plasmid success in a naturally evolved plasmid population.

I would find the manuscript overall somewhat easier to read if the experimental system would be presented at the results beginning with more detail. I understand that author's wish to connect this study to the previous manuscript, at the same time, it seems that they expect some prior knowledge of that work while reading the current study. A clearer presentation of the different plasmids and hosts in the beginning would help to understand the overall results much better.

Other general comments

General comment about the strain/clone naming: In line 102 the authors call one clone w_e, however afterwards they refer to it without further explanation by sw_e. I assume, that w_e and similar denominations refer to the plasmid itself, and sxyz to the bacteria carrying it. It would be helpful to make this clear in some part of the manuscript.

General comment about the figures displaying CFU/ml: I personally prefer the display of frequencies within the whole population or the recipient population, since life titer or recipient CFU/ml can change slightly daily and it is easier to comprehend the rarity of changes the population undergoes. In my opinion this would add another layer of understanding to the data graphs.

Comments on text and figures

Line 96-99: The seeding cell of the colony retained from the experiment must have harbored a mixture of R1 variants, I agree. However, resistance to ampicillin by bla is provided by secretion of the gene product. Therefore, the assumption of the plasmid variants co-existing within each bacterial clone is at least unprecise because segregational drift likely has led to homoplasmic daughter cells carrying one of either variants.

Line 109 - 111: Considering, that the authors base their latter detection method on the primer pair used here, I would like to see a deeper analysis of the faint band detected in the ancestral R1 carrying clone. This could either involve the sequencing of the PCR product or designing another primer set to validate that this is no PCR artefact from unspecific priming. Also, this faint band raises the question when colonies in latter experiments analyzed by colony PCR were considered carrying Rdel.

Line 136 - 142: I agree with the general conclusion that likely the Rdel variants also have a copA* allele encoded on them, while the AMR encoding plasmids still have the ancestral PCN. However, the combination of more variants (derived from homologous recombination between the two) could also be possible. The authors could consolidate their assumption by investigating the coverage of the copA* alleles vs. the coverage of copAWT in the same clone, then comparing it to the frequencies of the AMR coverage.

Figure 2: In Figure S2 the authors show that the strain carrying R1WT produces a faint band for the R1del primer pair as well. I assume the same strain has been used to start the evolution experiments. Therefore, showing the original R1del frequency in colonies similar to Fig. 2B or even better determining it by qPCR/ddPCR would allow for interpretation if new Rdel arose or if they have been already present in the starting culture. If latter is true, additionally segregational drift needs to be taken into account (assuming co-existence of plasmid variants in one cell) as well as fitness of homoplasmic carriers of one or the other variant (which would be best to be provided in any case, if possible). Unless this is cleared, I do not agree with the interpretation of "stealth plasmids evolve" (line 182).

Moreover, this analysis of t0 clones could also explain the differences between experiment 1 and experiment 2 (since the composition of plasmid variants could be different for the two starting cultures).

Line 190 following:

The results presented in Fig. 3 are tricky to interpret - the authors should supply data on plasmid stability (or segregational loss) in the form of % hosts in the population rather than frequency alone.

Furthermore, I cannot comprehend from the manuscript how the authors made sure their competition experiments were not influenced by horizontal transmission. Or at least that there is no plasmid loss during culturing, then it could be assumed that surface exclusion is preventing horizontal transmission. Since the authors show later, that transferring faster than R1, plasmid-free cells arising during the experiment could falsify results. In my opinion, either inhibition of conjugation e.g. by linolic acid (Ripoll-Rozada et al., 2016) or a stability experiment showing no substantial plasmid loss would be needed.

Line 196: There is no Figure 3B.

Line 207: Please add details on the design of that evolution experiment (beyond the caption of figure 4). Is that the same design as presented in Fig. 1A? What is the daily mating/ transfer regime?

Figure 4 and respective paragraphs: Since the horizontal spread of the plasmid over a prolonged time, bacterial growth i.e. inter-cellular competition could have an effect (as already observed for sw90d in Figure 3) as well as varying plasmid acquisition costs between R1 variants (see (Ahmad et al., 2023)).

This could be avoided by conducting the mating experiment and recording transconjugant frequencies already after a short time interval, that allowed for only 0-3 divisions would be more insightful.

Also - the figure will be easier to read with a proper legend (indeed, C is almost informative but 'none' = black? is 0 and hence not shown). A simple graphic illustration of the plasmid variants would be also very helpful.

Line 246: please elaborate on the motivation for this section. Is it possible that the results observed in Fig. 4 are due to fitness differences between hosts of the different plasmid variants? Answering that may require head-to-head competitions rather than growth rate comparisons.

Also - it would be easier to follow if the authors explain the two strains (MG ∆lac and MG RifR) earlier in the manuscript (I could not easily identify where are those coming from).

Line 283-285 - this conclusion seems to me as going beyond what can be interpreted from the comparison of growth rates without further validation, especially the AMR region deletion relieving the fitness cost. Such an assertion should be tested, eg, by replacing the AMR region by random DNA to restore the plasmid size. But maybe I'm missing something here.

Line 287 and Fig. 6 - this section is very interesting but I find it also somewhat misleading. The 'stealth plasmid' is a variant of R1, and the results show that all R1 variants are 'blocking' transfer. My conclusions would be that inhabitant plasmids (i.e., surface exclusion mechanism) block plasmid transfer better than defense mechanisms that operate upon invasion (and not as the authors state in the title of that section). This is somewhat expected. It would be good to add here a control with the 'wt' R1 - which should also block the transfer; i.e., this is probably not a unique aspect of the 'stealth' plasmid.

In that same section (Fig. 6) - I realize that the authors are testing for the plasmid presence consistently by the number of AMR CFUs, which was (if I get it right) the source of their 'blind spot' in the publication preceding this current work. Are those the AMR that are in the 'excised' island or other markers of the R1 backbone? If the plasmids they are tracking would experience a similar deletion, are the results still correct?

Line 308 - transfer would be more accurate here than 'spread'.

Line 336 following: I have trouble understanding how the high PCN has a direct effect on vertical transmission. As stated before, I would be careful to conclude this from their experiments without the respective controls. I do agree, that a higher PCN seems to have an effect on transmission (horizontally and maybe vertically). But I am wondering which reasoning could there be for the vertical transmission advantage. As far as I understand (Brantl, 2007; Lilly & Camps, 2015) the effect of one copA* mutant would not affect the respective plasmid carrying the allele but all plasmid variants present in the cell's plasmid pool.

I have no comments on the discussion as the results feel rather 'unstable' to me hence the conclusions may change.

The methods are well described.

References

Zhang, X., Deatherage, D.E., Zheng, H. et al. Evolution of satellite plasmids can prolong the maintenance of newly acquired accessory genes in bacteria. Nat Commun 10, 5809 (2019). https://doi.org/10.1038/s41467-019-13709-x

Ahmad, M., Prensky, H., Balestrieri, J., ElNaggar, S., Gomez-Simmonds, A., Uhlemann, A.-C., Traxler, B., Singh, A., & Lopatkin, A. J. (2023). Tradeoff between lag time and growth rate drives the plasmid acquisition cost. Nature Communications, 14(1), 2343. https://doi.org/10.1038/s41467-023-38022-6

Brantl, S. (2007). Regulatory mechanisms employed by cis-encoded antisense RNAs. Current Opinion in Microbiology, 10(2), 102-109. https://doi.org/10.1016/j.mib.2007.03.012

Dimitriu, T., Matthews, A. C., & Buckling, A. (2021). Increased copy number couples the evolution of plasmid horizontal transmission and plasmid-encoded antibiotic resistance. Proceedings of the National Academy of Sciences, 118(31), e2107818118. https://doi.org/10.1073/pnas.2107818118

Lilly, J., & Camps, M. (2015). Mechanisms of Theta Plasmid Replication. Microbiology Spectrum, 3(1), 3.1.02. https://doi.org/10.1128/microbiolspec.PLAS-0029-2014

Ripoll-Rozada, J., García-Cazorla, Y., Getino, M., Machón, C., Sanabria-Ríos, D., de la Cruz, F., Cabezón, E., & Arechaga, I. (2016). Type IV traffic ATPase TrwD as molecular target to inhibit bacterial conjugation. Molecular Microbiology, 100(5), 912-921. https://doi.org/10.1111/mmi.13359

---

## [Decision Letter · Decision Letter 2]

20 Nov 2025

Dear Dr Dimitriu,

Thank you for your patience while we considered your revised manuscript "Plasmid streamlining drives the extinction of antibiotic resistance plasmids under selection for horizontal transmission" for consideration as a Research Article at PLOS Biology. Please accept my sincere apologies for the delays that you have experienced during this round of the peer review process. Your revised study has now been evaluated by the PLOS Biology editors, the Academic Editor and two of the original reviewers.

In light of the reviews, which you will find at the end of this email, we are pleased to offer you the opportunity to address the remaining points from the reviewers in a revision that we anticipate should not take you very long. After discussions with the Academic Editor, while we agree that the suggestion by Reviewer #1 to provide additional PCR data to remedy PCR bias in the detection of the wildtype and streamlined plasmid would be useful, we think this can be optional for the revision. We will then assess your revised manuscript and your response to the reviewers' comments with our Academic Editor aiming to avoid further rounds of peer-review, although we might need to consult with the reviewers, depending on the nature of the revisions.

-IMPORTANT-

In addition, I would be grateful if you could please address the following editorial and data-related requests that I have provided below (A-E):

(A) Please also ensure that each of the relevant figure legends in your manuscript include information on *WHERE THE UNDERLYING DATA CAN BE FOUND*, and ensure your supplemental data file/s has a legend (i.e. 'The underlying data can be found in S1_Data').

(B) We require the original, uncropped and minimally adjusted images supporting all blot and gel results reported in the following Figures:

Figure S2A-D, S4A-D

We will require these files before a manuscript can be accepted so please prepare and upload them now. Please carefully read our guidelines for how to prepare and upload this data: https://journals.plos.org/plosbiology/s/figures#loc-blot-and-gel-reporting-requirements.

(C) Per journal policy, if you have generated any custom code during the course of this investigation, please make it available without restrictions. Please ensure that the code is sufficiently well documented and reusable, and that your Data Statement in the Editorial Manager submission system accurately describes where your code can be found.

(D) Please ensure that your Data Statement in the submission system accurately describes where your data can be found and is in final format, as it will be published as written there.

(E) Please note that per journal policy, the model system/species studied should be clearly stated in the abstract of your manuscript.

**IMPORTANT - SUBMITTING YOUR REVISION**

*Resubmission Checklist*

*Published Peer Review*

*PLOS Data Policy*

*Blot and Gel Data Policy*

Best regards,

Richard

Richard Hodge, PhD

rhodge@plos.org

REVIEWS:

Reviewer #1: Overall, I am mostly satisfied with the revisions the authors have made to the manuscript. I am happy to see a model that helps to generalise and extend the main findings, and highlights some interesting areas for follow-up studies. There remain some (relatively minor) outstanding issues, as I indicate below.

- Line 119. Clarify. In the text above "isolated one plasmid clone" but here referring to "populations" again. May need a few words just to clarify, e.g. "heterogeneity in the population emerging from the single clone"

- Line 120ff. I wonder if this section could be clarified further. "We took advantage of control populations from our evolution experiment, which had been established without any plasmids, and for which we had sequenced clones without any antibiotic selection step". "Plasmid-free" is not correct, as you describe a plasmid in this population.

- Line 136ff dissects the PCR methodology to identify streamlined plasmids. In my opinion, and in the context of this paper, extended dissection of these PCR artefacts seems to be a digression into technical curiosities (i.e. template switching) rather than relevant results, and might be better placed in the methods or supplemental information.

- I agree with the authors that there are visible differences in band intensity, and even though the 'faint band' is a bit too intense for my liking (particularly in S2D top row), the controls included in these experiments means that I trust the results shown e.g. in Fig. 2. The only issue comes when there are heterotypic cells containing both plasmid variants, there might be a spectrum of intensities making it difficult to distinguish low-frequency deletants from the spurious signal from full length plasmids. The authors should ideally validate key results in Fig 2 by repeating a subset with the longer, more stringent primer pairs (though I note a faint band in the second R1 lane in S2D bottom row...). I also wonder if the authors could develop an alternative assay whereby the reverse primer is located within the deleted region (thus giving a band for the non-streamlined plasmid) which might be more robust to template switching. A design might be implemented whereby three primers are used, with a common 'outside' primer, an AMR-internal-region primer that generates a smaller product for the wild-type plasmid, and a primer at the other end of the deleted region that gives a larger product for the streamlined plasmid. This design would likewise suffer from low sensitivity at detecting the deletants. However, one advantage would be that for clones dominated by full-length plasmids, competition between amplicons (favouring shorter amplicons) would mean that full-length plasmid clones would generate primarily/only the shorter amplicons, titrating out the low-frequency artefacts, and giving a strong positive signal for the continued presence of the full-length plasmid in the tested population. This may be excessive for the current manuscript but could be considered in future studies.

- Line 139. The authors might like to indicate here that the alternative hypothesis — that the variant emerged by transposase-mediated deletion between the outer repeats of the IS1 copies — is not supported by the retention of a copy of IS1.

- The mathematical model is a nice addition to the paper and highlights the importance of plasmid-plasmid competition in the spread of traits. The authors might like to make it clearer that the role of streamlined plasmids in helping cells resist AMR plasmids is premised on the infectious transmission of AMR, rather than under selective conditions. For example, in the abstract line 86. Next, it would be interesting to look at how streamlined plasmids affect AMR invasion under conditions where selection for full-length plasmids is applied, or applied in a fluctuating or spatial-structured manner… (I'm sure the authors have this idea in mind already!)

- Line 232 — suggest reword: "are expected to be primarily driven by competition rather than evolutionary dynamics." Evolutionary dynamics can still play an important role in the short term.

- It is perhaps a bit unconventional for gamma to indicate death in the model, rather than conjugation, but it's clearly signposted, so I don't mind too much.

- Important to define 'vertical transmission'. Vertical transmission often taken to include plasmid cost, but the distinction is made here such that 'vertical transmission' refers mainly to persisting in the face of (or outcompeting) co-resident plasmids, i.e. segregational effects (line 472).

Reviewer #2: The manuscript 'Plasmid streamlining drives the extinction of antibiotic resistance plasmids under selection for horizontal transmission' by Matthews, Lehtinen & Dimitriu is now much clearer compared to the previous version. I commend the authors for their efforts in improving the clarity and presentation.

The authors have addressed all of my previous questions and have revised the manuscript accordingly. In my view, they have also responded thoroughly to the major suggestions and criticisms raised by the other two reviewers.

I have only a few minor comments:

(Lines of the new version - the one without track-changes)

Introduction, Lines 68-70, I'd suggest joining these two sentences. Something like: "We previously studied experimentally the short-term evolution of R1, a model conjugative plasmid conferring resistance to multiple antibiotics, and one of the first and best studied conjugative plasmids (16,17). "

Results section, line 110: I could not understand this: "dropped to values either higher than the ancestral plasmid, …".

Legend of Fig. 1: I'd suggest including some explanations here: the meaning of m, w, and the treatments

Several places (e.g. line 477): β-lactamases instead of B-lactamases

Methods: how much higher is the mutation rate of mutL mutants? 50x?

Line 573: " Iin the experiments shown in Fig. 6…" should be: "In the experiments shown in Fig. 6…"

---

## [Editor Report · Decision Letter 3]

28 Nov 2025

Dear Tatiana,

On behalf of my colleagues and the Academic Editor, Arjan de Visser, I am pleased to say that we can accept your manuscript for publication, provided you address any remaining formatting and reporting issues. These will be detailed in an email you should receive within 2-3 business days from our colleagues in the journal operations team; no action is required from you until then. Please note that we will not be able to formally accept your manuscript and schedule it for publication until you have completed any requested changes.

PRESS

Best wishes, 

Richard

Richard Hodge, PhD

rhodge@plos.org

PLOS
